# Factors Associated with Daily Fruit and Vegetable Intakes among Children Aged 1–5 Years in the United States

**DOI:** 10.3390/nu16050751

**Published:** 2024-03-06

**Authors:** Adi Noiman, Seung Hee Lee, Kristin J. Marks, Mary Ellen Grap, Carrie Dooyema, Heather C. Hamner

**Affiliations:** 1Division of Nutrition, Physical Activity, and Obesity, National Center for Chronic Disease Prevention and Health Promotion, Centers for Disease Control and Prevention, Atlanta, GA 30341, USA; xde5@cdc.gov (S.H.L.); kma8@cdc.gov (K.J.M.); uco3@cdc.gov (M.E.G.); igb7@cdc.gov (C.D.); hfc2@cdc.gov (H.C.H.); 2Epidemic Intelligence Service, Centers for Disease Control and Prevention, Atlanta, GA 30341, USA; 3US Public Health Service Commissioned Corps, Rockville, MD 20852, USA; 4Oak Ridge Institute for Science and Education, Oak Ridge, TN 37830, USA

**Keywords:** early childhood nutrition, fruit and vegetable intakes, multiple logistic regression

## Abstract

To describe child, caregiver, and household characteristics associated with fruit and vegetable intakes among US children aged 1–5 years, we examined fruit and vegetable intakes (less than daily vs. daily) using data from the 2021 National Survey of Children’s Health among children aged 1–5 years. Multiple logistic regression provided adjusted odds ratios for factors associated with (1) daily fruit and (2) daily vegetable intakes. Among children aged 1–5 years, 68% (n = 11,124) consumed fruit daily, and 51% (n = 8292) consumed vegetables daily. Both daily fruit and daily vegetable intake were associated with child age, child race and ethnicity, and frequency of family meals. For example, children who ate a family meal 4–6 days/week (aOR 0.69; 95% CI 0.57, 0.83) or 0–3 days/week (aOR 0.57; 95% CI 0.46, 0.72) were less likely to consume fruit daily compared to children who had a family meal every day. Participation in food assistance programs, food insufficiency, and household income were not significantly associated with odds of daily fruit or daily vegetable intake in the adjusted models. Several factors were associated with daily fruit and vegetable intake among children aged 1–5. Strategies aimed at increasing fruit and vegetable consumption in early childhood may consider these child, caregiver, and household characteristics. Pediatric healthcare providers, early childhood education centers, and families of young children may be important partners in this work.

## 1. Introduction

Nutrient-dense diets support optimal growth, health, and cognitive development for young children, and they are important in the first 5 years of life [1]. Fruits and vegetables can provide many of these nutrients, including fiber, folate, potassium, vitamin A, vitamin C, and vitamin K [2]. Dietary patterns established in early life may influence diets in adolescence and adulthood [3], and diets rich in fruits and vegetables are associated with positive health outcomes for children and adults, including reduced risks for developing high blood pressure, heart disease, type 2 diabetes, and some cancers [4,5,6]. In addition, the first 1000 days of life are crucial for brain development, and a lack of key nutrients during this period may lead to lifelong deficits in brain function that cannot be overcome [1]. The health benefits associated with consuming fruits and vegetables, coupled with findings that dietary patterns develop at an early age, support the need to promote fruit and vegetable consumption in early childhood. Nevertheless, national data indicate that, on average, children of all ages are not eating the recommended amounts of fruits or vegetables [7]. Specifically, a report using data from the National Survey of Children’s Health (NSCH) found that, in 2021, one-third of children aged 1–5 years did not eat fruit daily in the preceding week, and one-half did not eat vegetables daily [8].

Limited information exists about the factors associated with fruit and vegetable intake among children under 5 years of age. A recent scoping review of the studies conducted in high-income countries summarized child, parent, household, and childcare level influences on preschool children’s dietary intake, as well as external factors such as increased exposure through various forms of media and play, nonfood rewards, and food manipulation (e.g., portion size, flavor, and visual appearance) [9]. However, outcome measures varied greatly by study, and much of the work was limited by relatively small sample sizes or a lack of national representativeness. The NSCH report used nationally representative data and found that daily fruit and vegetable intake was lower among children who were aged 2–5 years, Black, or lived in households with limited food sufficiency, but it did not account for potential confounding [8]. This analysis identified child, caregiver, and household characteristics associated with fruit and vegetable intake among a nationally representative sample of children aged 1–5 years in the United States.

## 2. Methods

### 2.1. The National Survey of Children’s Health, 2021

The NSCH is an annual, nationally representative survey of noninstitutionalized children aged 0 to 17 years, funded by the Health Resources and Services Administration (HRSA) Maternal and Child Health Bureau (MCHB) and conducted by the US Census Bureau. The survey provides national and state estimates for a variety of child health indicators. The NSCH Methodology Report provides a detailed description of the NSCH sampling and data collection procedures [10]. Briefly, potential respondents were mailed an invitation to complete a screening questionnaire and asked if one or more children between the ages of 0 and 17 live in the household. Eligible respondents completed the screening questionnaire about all children in the household, and one child from each household was randomly selected. Survey responses were collected online, by mail, or by phone, in English or Spanish. In the 2021 survey, children under 5 years of age were oversampled to ensure robust data estimates for this age group. The NSCH provides sample weights for population-based estimates to account for potential bias due to nonresponse or selection procedures. Poststratification adjustment was used to ensure that sociodemographic subgroups were appropriately represented in the estimates.

### 2.2. The Analytic Sample

This analysis used 2021 survey data, the first survey year to include questions about fruit and vegetable consumption; questions were limited to children aged 1 to 5 years. Data were collected between June 2021 and January 2022. The 2021 weighted overall response and interview completion rates were 40.3% and 79.5%, respectively [10]. There were 18,830 children aged 1 to 5 years, and children were excluded if they were missing values for fruit or vegetable intake (n = 412), resulting in an analytic sample of 18,418.

### 2.3. Outcomes

Daily fruit and vegetable intakes were the two outcomes of interest. Respondents were asked, during the past week (1) how many times did this child eat fruit? and (2) how many times did this child eat vegetables? Response options included 0 times per week, 1–3 times per week, 4–6 times per week, 1 time per day, 2 times per day, or 3 or more times per day in the last week. We categorized each outcome as less than daily (0, 1–3 times, and 4–6 times during the past week) or daily (1 time, 2 times, and 3 or more times per day) in the last week. Respondents were asked to include fresh, frozen, canned, or dried fruits and vegetables in their responses and to exclude juice, french fries, fried potatoes, or potato chips.

### 2.4. Exposures

Exposures included child, caregiver, and household characteristics. Child characteristics included age (1–5 years), sex (male or female), and caregiver-reported child race and ethnicity. The race and ethnicity categories included Asian, non-Hispanic (“Asian”); Black or African American, non-Hispanic (“Black”); Hispanic or Latino (“Hispanic”); multiracial or other race, non-Hispanic (“other/multiracial”); or White, non-Hispanic (“White”). The race and ethnicity indicator was included to account for the potential effect of cultural differences on childhood diet.

Caregiver characteristics included maternal age (≤25 or >25 years), highest caregiver education level (≤high school, any college, or ≥college), and childcare received for ≥10 h per week from someone other than the parent or guardian (yes or no).

Household characteristics included the total number of children (≤2 or >2); metropolitan statistical area (MSA) status, defined as having ≥1 urbanized area with a population of ≥50,000 [11] (yes or no); and income. The NSCH reports income as a percentage of the federal poverty level (FPL) by family composition. We categorized income as <130%, 130–349%, or ≥350% FPL. FPL missing values were multiply-imputed according to the NSCH guidelines [12].

Additional household characteristics included food assistance program participation, food insufficiency, and frequency of family meals. We defined food assistance program participation as participation in one or more programs in the last 12 months, including the Supplemental Nutrition Assistance Program (SNAP); free or reduced-cost school meals; or the Special Supplemental Nutrition Program for Women, Infants, and Children (WIC) (yes or no). Food insufficiency was assessed by asking “Which of the following best describes your household’s ability to afford the food you need during the past 12 months?” We categorized responses into “high” (sometimes or often could not afford enough to eat), “marginal” (could always afford enough to eat but not always the kinds of foods we should eat), or “low” (could always afford to eat good nutritious meals). The frequency of family meals was estimated by asking “During the past week, on how many days did all the family members who live in the household eat a meal together?” We categorized responses into 0–3 days, 4–6 days, or every day in the past week.

### 2.5. Statistical Analysis

We calculated the percent of children consuming (1) daily fruits and (2) daily vegetables overall and by each exposure. Differences in daily intakes were assessed using chi-square tests. Multiple logistic regression was performed to estimate adjusted odds ratios (aORs) and 95% confidence intervals (CIs) for factors associated with (1) daily fruit intake and (2) daily vegetable intake. All exposures with significant chi-square results (*p* < 0.05) were included in the models. For each included exposure, the category with the largest proportion of reported daily intake was the reference group. Because the NSCH uses sample weights, we present unweighted sample sizes and weighted estimates (percentages, aORs). SAS-callable SUDAAN version 11.0 was used to account for the complex survey design [12].

Because socioeconomic exposures (household income, food assistance program participation, and food insufficiency) might be highly correlated, we assessed collinearity using Cramer’s V statistic. A Cramer’s V of >0.5 was considered a strong association. We also performed a sensitivity analysis that compared the results from our fully adjusted models to the results from adjusted models, including only one socioeconomic exposure at a time.

## 3. Results

This analysis included 18,418 children aged 1–5 years. The age and sex distributions were approximately equal, with 20% and 50% of children in each age and sex group, respectively. Approximately 5% of children were Asian (n = 1052), 7% were multiracial or another race (n = 1568), 13% were Black (n = 1066), 25% were Hispanic (n = 2415), and 50% were non-Hispanic White (n = 12,317). Overall, 68% (n = 11,124) and 51% (n = 8292) consumed fruit or vegetables daily, respectively.

### 3.1. Daily Fruit Intake

The percentage of children consuming daily fruit was higher among children aged 1 year (75%) compared to those aged 5 years (64%), and a larger percentage of children aged 5 years consumed no fruit in the last week (1.9%), compared to those aged 1 year (0.8%) (Figure 1). Daily fruit intake was significantly different across all child, caregiver, and household characteristics, except for child sex, number of children in the household, and MSA status (Table 1). After adjustment, the factors significantly associated with daily fruit intake included child age, child race and ethnicity, maternal age, caregiver education, and frequency of family meals (Table 2). Children who were older (2, 4, or 5 years vs. 1 year), Black or Asian (vs. White), with younger mothers (<25 vs. ≥25), less educated caregivers (some college vs. college), or from households eating less frequent family meals per week (4–6 days/week or 0–3 days/week vs. every day) had significantly decreased odds of consuming daily fruit. For example, children who ate a family meal 4–6 days/week or 0–3 days/week were less likely to consume daily fruit compared to children who had a family meal every day (aOR 0.69; 95% CI 0.57, 0.83; aOR 0.57; 95% CI 0.46, 0.72, respectively).

### 3.2. Daily Vegetable Intake

Daily vegetable intake was less prevalent than daily fruit intake, overall and across exposures (Table 1). The percentage consuming daily vegetables was higher among those aged 1 year (56%) than among those aged 5 years (47%), and a larger percentage of children aged 5 years consumed no vegetables in the last week (8.3%), compared to those aged 1 year (2.8%) (Figure 2). Daily vegetable intake was significantly different across all child, caregiver, and household characteristics, except for child sex, number of children in the household, and MSA status (Table 1). After adjustment, factors significantly associated with daily vegetable intake included child age, child race and ethnicity, childcare from someone other than the parent or guardian, and the frequency of family meals (Table 2). Children who were older (5 years vs. 1 year), Black or Hispanic (vs. White), received <10 h per week of childcare from someone other than the parent or guardian (vs. ≥10 h), or from households eating less frequent family meals per week (4–6 days/week or 0–3 days/week vs. every day) had significantly decreased odds of consuming daily vegetables. For example, children who were Black or Hispanic were less likely to consume daily vegetables than children who were White (aOR 0.51; 95% CI 0.38, 0.67; aOR: 0.70; 95% CI 0.57, 0.87, respectively).

### 3.3. Collinearity

The Cramer’s V statistic was <0.5 between each pair of socioeconomic exposures. Our sensitivity analysis yielded similar results between the adjusted models, including only one socioeconomic exposure at a time and the fully adjusted models, indicating that collinearity was unlikely.

## 4. Discussion

We examined child, caregiver, and household characteristics associated with daily fruit and vegetable intakes among children 1 to 5 years of age in the United States. Children who were aged ≥1 year, Black, or from households eating fewer family meals per week were significantly less likely to consume daily fruits and vegetables. Socioeconomic factors, including household income, participation in food assistance programs, and food insufficiency in the past 12 months, were not significantly associated with daily fruit or vegetable intake after adjustment.

Our findings indicating that children aged ≥1 year experienced decreased odds of consuming daily fruits and vegetables align with previous studies that found diet quality, including fruits and vegetables, tends to decrease with increasing age [13]. Research shows that this trend continues into adolescence [14,15]. There is also evidence suggesting that infrequent intake of fruits and vegetables in late infancy is associated with infrequent intake at 6 years of age [16] and that food preferences and eating habits established in childhood and adolescence continue in adulthood [17]. Together with our results, these findings highlight the importance of early guidance about fruit and vegetable consumption for families of young children. Providers can take advantage of frequent well-child visits in early childhood to raise awareness about low rates of fruit and vegetable consumption and the importance of encouraging daily fruit and vegetable intake for optimal growth and health outcomes. The American Academy of Pediatrics (AAP) Bright Futures Nutrition 3rd Edition Pocket Guide provides health professionals with discussion topics and interview questions related to child diet to use with families during well-child visits [16], and several tools have been developed for clinicians by the AAP’s Institute for Healthy Childhood Weight to promote healthy childhood nutrition [18]. Examples include clinical education opportunities like the “Early Child Nutrition: From Birth to Two Years” webinar [19] and technology platforms that generate personalized patient education materials like the Healthy Growth app [20].

Approximately 12.5 million children under 5 years of age receive nonparental care at least once per week in the United States [21]. Our study found that children receiving childcare from someone other than the parent or guardian for ≥10 h per week were more likely to consume a daily vegetable than those receiving <10 h, highlighting an opportunity to explore the impact of early childhood education (ECE) settings on eating behaviors of young children. A systematic review identified nutrition education as one effective ECE intervention to increase fruit and vegetable intakes [22], but only six studies were included in the review. ECE policies and activities are a priority strategy for the CDC to improve early childhood nutrition, including farm-to-ECE activities like gardening, farm visits, and tasting fresh produce from local farms [23]. A randomized controlled trial assessing the impact of a farm-to-school nutrition and gardening intervention for Native American families found a significantly positive association between the intervention and vegetable intake among children [24].

Like our study, previous research has found that non-Hispanic Black children are less likely than non-Hispanic White children to consume fruits [25] and vegetables [26]. However, the literature includes inconsistent findings regarding the relationship between race and ethnicity and fruit and vegetable consumption among children of other racial and ethnic groups [27]. More work is needed, including qualitative work in consultation with families and communities, to better understand the barriers, facilitators, and cultural differences affecting fruit and vegetable intake in early childhood. For instance, findings from one study suggest differences in maternal recall of healthcare provider recommendations for fruit and vegetable intake by race and ethnicity [28]. Among mothers of children aged 6 months to 5 years, those who identified as Hispanic were significantly less likely to recall being advised to offer a variety of fruits and vegetables to their child compared to those who identified as White [29].

Similarly, efforts to support culturally appropriate nutrition education and experiences in ECE settings could encourage more children to eat fruits and vegetables. Indeed, a position paper published by the Academy of Nutrition and Dietetics recommends that ECE programs provide a variety of healthy foods reflecting the cultural preferences of families to effectively meet the children’s nutrition needs [30]. The SNAP-education “Edible ABC’s” curriculum, which includes picture cards, videos, and a “tasting at home” family newsletter [31], is one initiative that could be adapted to reflect the cultures and languages of the communities accessing care.

Our analysis found that the frequency of family meals was positively associated with daily fruit and vegetable intake regardless of household income, participation in food assistance programs, or food insufficiency, signifying that families from all socioeconomic backgrounds may benefit from sharing meals. Past research supports this finding [32,33], suggesting that interventions helping families increase the number of family meals per week could lead to more children eating fruits and vegetables. Several pediatric organizations and healthcare institutions recommend family meals to improve child nutrition and eating behaviors [34,35]; however, a survey of AAP members found that while almost 90% of pediatricians discussed consuming daily fruits and vegetables with parents of children under age 2 years, less than half discussed eating meals together as a family [36]. Incorporating messaging about family meals into anticipatory guidance provided during well-child visits could potentially improve rates of daily fruit and vegetable intake for young children.

More work is needed to understand which characteristics of family meals are most important for promoting fruit and vegetable intake. For example, a randomized controlled trial found that longer meal times significantly increased the number of fruits and vegetables consumed by children during the meal [37]. Nevertheless, previous research shows that parents’ eating behaviors and feeding practices are linked to children’s eating behaviors [38], so family meals may provide time for parents and caregivers to model healthy eating behaviors.

In contrast to some previously published research [27,39], we did not find a significant association between daily fruit and vegetable intake and socioeconomic factors, such as household income, participation in food assistance programs, or food insufficiency, after adjustment. However, there is support for our findings in the literature. A mediation analysis measuring the association between socioeconomic factors, parental role modeling, and children’s fruit and vegetable consumption found that family income was not directly associated with fruit or vegetable consumption but was indirectly predictive through more frequent parent role modeling of fruit and vegetable consumption [40]. The frequency of family meals may mediate the relationship between socioeconomic status and fruit and vegetable intake, as family meals may be an opportunity for caregivers to model healthy eating behaviors, but more research is needed.

This work is novel because we analyzed fruit and vegetable intake with questions newly added to the NSCH. The data are nationally representative and include a robust sample of children aged 1 through 5 years, an under-represented age group in the existing literature on fruit and vegetable consumption. This analysis also has limitations. First, the data are cross-sectional, and causality cannot be inferred. Second, fruit and vegetable intakes reported by caregivers could be subject to recall or social desirability bias, and caregivers might not be aware of food consumed outside the home. Third, no survey questions addressed the types of fruit and vegetables consumed (e.g., leafy greens vs. starchy vegetables). Similarly, the survey questions do not allow respondents to indicate if the consumed canned or frozen fruits and vegetables had added sugar, syrup, or sauces that could change the nutritional content. Finally, the NSCH measured frequencies but not quantities, so findings cannot be used to assess whether children are meeting the national fruit and vegetable recommendations.

## 5. Conclusions

Using recent, nationally representative data collected about children aged 1–5 years in the United States, we found that child age, child race and ethnicity, and the frequency of family meals were associated with daily fruit and vegetable intakes, after adjustment. Strategies to promote fruit and vegetable consumption among young children could consider these factors to improve intake and support optimal growth and development. Interventions targeting people with whom children regularly interact and places where young children regularly spend time, such as families, healthcare providers, and ECE settings, could help to increase daily fruit and vegetable intake in early childhood.

## Figures and Tables

**Figure 1 nutrients-16-00751-f001:**
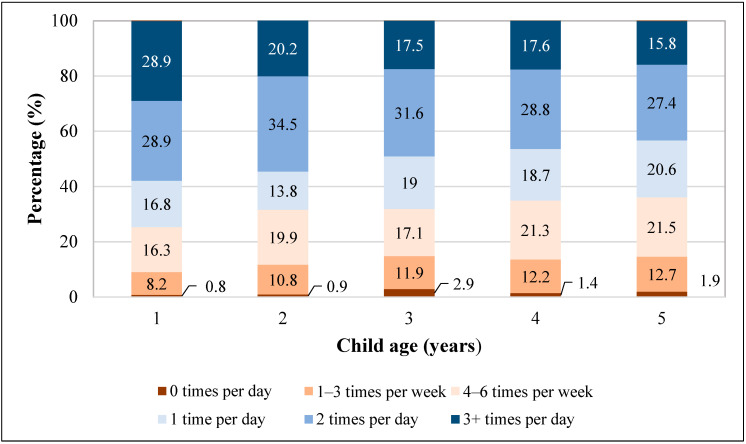
Fruit intake frequencies among children aged 1–5 years, National Survey of Children’s Health, 2021 *. * Percentages are weighted to account for complex survey design.

**Figure 2 nutrients-16-00751-f002:**
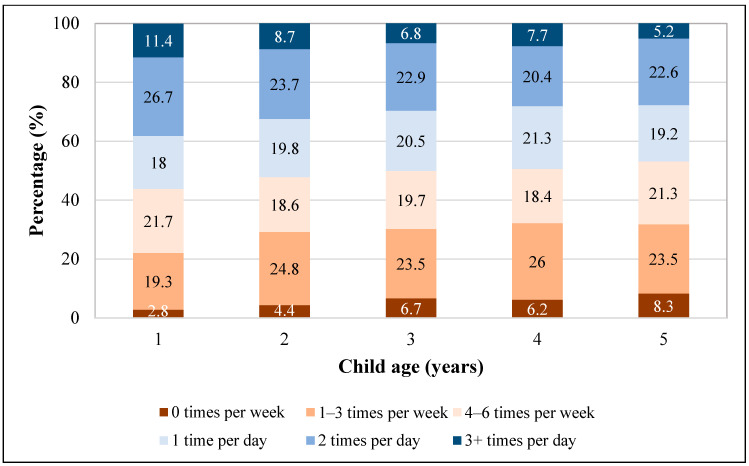
Vegetable intake frequencies among children aged 1–5 years, National Survey of Children’s Health, 2021 *. * Percentages are weighted to account for complex survey design.

**Table 1 nutrients-16-00751-t001:** Reported daily fruit and vegetable intakes for children aged 1–5 by child, caregiver, and household characteristics, National Survey of Children’s Health, 2021.

Characteristics	Total n (%) ^a^	Daily Intakes			
		Fruits ^b^ % (95% CI) ^a^	*p*-Value ^‡^	Vegetables ^c^ % (95% CI) ^a^	*p*-Value ^‡^
**Total (N = 18,418)** ^**c**^		(n = 11,124) 67.9 (66.2, 69.6)		(n = 8292) 50.9 (49.2, 52.7)	
	**Child Characteristics**				
**Age** (years)			0.002		0.03
1	2439 (18.9)	74.7 (70.4, 78.5)		56.2 (51.6, 60.6)	
2	4233 (19.9)	68.4 (64.9, 71.7)		52.3 (48.8, 55.7)	
3	3807 (20.3)	68.1 (64.4, 71.5)		50.2 (46.2, 54.0)	
4	3981 (21.3)	65.1 (61.0, 68.9)		49.4 (45.3, 53.5)	
5	3958 (19.5)	63.9 (60.2, 67.4)		47.0 (43.2, 50.8)	
**Sex**			0.72		0.05
Male	9566 (51.2)	68.2 (65.9, 70.5)		49.2 (46.7, 51.6)	
Female	8852 (48.8)	67.6 (65.1, 70.0)		52.8 (50.2, 55.3)	
**Race and ethnicity**			<0.001		<0.001
Asian, non-Hispanic	1052 (5.0)	57.8 (50.2, 65.1)		52.5 (45.2, 59.6)	
Black, non-Hispanic	1066 (13.0)	49.4 (43.8, 55.0)		35.4 (30.2, 40.9)	
Hispanic	2415 (25.3)	67.9 (66.2, 69.6)		46.3 (41.5, 51.1)	
Other/multiracial, non-Hispanic ^d^	1568 (7.0)	73.2 (68.5, 77.4)		55.9 (50.7, 61.0)	
White, non-Hispanic	12,317 (49.7)	73.8 (72.2, 75.3)		56.5 (54.7, 58.3)	
	**Caregiver Characteristics**				
**Maternal age (years)**			<0.001		0.002
>25	15,199 (79.7)	70.6 (68.8, 72.3)		52.5 (50.5, 54.4)	
≤25	3219 (20.3)	57.5 (53.1, 61.8)		44.9 (40.8, 49.2)	
**Caregiver education**			<0.001		<0.001
≥College degree	12,392 (54.9)	74.7 (72.9, 76.5)		54.9 (52.9, 57.0)	
Some college or Associate’s	5666 (36.7)	59.0 (56.0, 61.9)		44.5 (41.6, 47.5)	
≤High school	360 (8.4)	62.7 (52.9, 71.6)		52.9 (42.9, 62.6)	
**Childcare from someone other than parent or guardian for ≥10 h per week**			<0.001		0.01
Yes	10,704 (49.7)	70.9 (68.9, 72.9)		53.1 (50.9, 55.4)	
No	7460 (50.3)	65.1 (62.4, 67.8)		48.6 (45.8, 51.3)	
	**Household Characteristics**				
**No. of children in the household**			0.17		0.15
≤2	14,860 (64.8)	68.9 (66.9, 70.7)		52.0 (50.0, 54.0)	
>2	3558 (35.2)	66.2 (63.0, 69.3)		49.0 (45.6, 52.5)	
**Total (N = 18,418) ^c^**		(n = 11,124) 67.9 (66.2, 69.6)		(n = 8292) 50.9 (49.2, 52.7)	
**MSA Status** (n = 16,507)			0.06		0.56
Yes	15,456 (88.2)	68.1 (66.1, 69.9)		50.6 (48.6, 52.7)	
No	2962 (11.8)	63.7 (59.6, 67.7)		52.0 (47.9, 56.0)	
**Household income ^e^**			<0.001		<0.001
<130% FPL	2919 (25.1)	60.6 (56.5, 64.6)		45.4 (41.2, 49.7)	
130% to <350% FPL	6350 (36.5)	65.1 (62.1, 67.9)		49.6 (46.7, 52.5)	
≥350% FPL	9149 (38.4)	75.4 (73.3, 77.5)		55.8 (53.4, 58.2)	
**Food assistance in the past 12 months ^f^** (n = 18,001)			<0.001		<0.001
Yes	5116 (40.8)	60.6 (57.5, 63.7)		45.8 (42.6, 49.1)	
No	12,885 (59.2)	73.7 (71.9, 75.4)		54.7 (52.7, 56.7)	
**Food insufficiency the past 12 months ^g^** (n = 18,075)			<0.001		<0.001
Low	14,501 (75.3)	70.4 (68.5, 72.3)		53.5 (51.5, 55.5)	
Marginal	3225 (21.6)	63.0 (59.2, 66.7)		43.9 (39.7, 48.1)	
High	349 (2.6)	53.7 (43.4, 63.7)		41.0 (31.1, 51.8)	
**No. family meals in the past week** (n = 18,183)			<0.001		<0.001
0–3 day	2746 (17.0)	58.2 (53.8, 62.6)		35.3 (31.2, 39.5)	
4–6 days	4400 (22.5)	66.4 (63.0, 69.7)		48.8 (45.2, 52.4)	
Every day	11,037 (60.5)	71.6 (69.4, 73.7)		56.2 (53.9, 58.5)	

Abbreviations: CI, confidence interval; MSA, metropolitan statistical area; FPL, federal poverty level. ^‡^ χ^2^ tests were used for each variable to examine differences across categories, and *p* < 0.05 was considered statistically significant. ^a^ Counts represent total numbers (unweighted); percentages are weighted to account for complex survey design. ^b^ Daily fruit was assessed by asking “During the past week, how many times did this child eat fruit? *Include any that were fresh, frozen, canned, or dried. Do not include juice.*” “Daily” was defined as responses of either 1 time, 2 times, or 3 or more times per day in the last week. ^c^ Daily vegetable was assessed by asking “During the past week, how many times did this child eat vegetables?” *Include any that were fresh, frozen, or canned. Do not include french fries, fried potatoes, or potato chips*.” “Daily” was defined as responses of either 1 time, 2 times, or 3 or more times per day in the last week. ^d^ This variable includes non-Hispanic American Indian or Alaska Native, Native Hawaiian and Other Pacific Islander, or Multiracial. ^e^ This variable represents household income as a percentage of the federal poverty level (FPL) by family composition. This variable was multiply-imputed according to NSCH guidelines to account for missing values. Categories reflect cut points for participation in SNAP and provide relatively equal sample sizes for each income group. The 2021 federal poverty guidelines can be found at https://aspe.hhs.gov/2021-poverty-guidelines#guidelines (accessed on 19 October 2023). ^f^ This variable reflects participation in SNAP or WIC or Free/Reduced Lunch benefits in the previous 12 months. ^g^ Food insufficiency was defined as “low” (could always afford to eat good nutritious meals), “marginal” (could always afford enough to eat but not always the kinds of foods we should eat), or “high” (sometimes or often could not afford enough to eat).

**Table 2 nutrients-16-00751-t002:** Factors associated with daily fruit and vegetable intakes among children aged 1–5, National Survey of Children’s Health, 2021 ^‡^.

Characteristics ^a^	Daily Fruit Intake ^b^ aOR (95% CI)	Daily Vegetable Intake ^c^ aOR (95%CI)
**Child Characteristics**		
**Age** (years)		
1	Reference	Reference
2	**0.70 (0.53, 0.92)**	0.79 (0.62, 1.00)
3	0.76 (0.57, 1.00)	0.79 (0.62, 1.01)
4	**0.65 (0.49, 0.87)**	0.77 (0.60, 1.00)
5	**0.59 (0.45, 0.79)**	**0.67 (0.53, 0.86)**
**Race and ethnicity**		
White, non-Hispanic	Reference	Reference
Asian, non-Hispanic	**0.44 (0.32, 0.61)**	0.78 (0.58, 1.07)
Black, non-Hispanic	**0.46 (0.36, 0.61)**	**0.51 (0.38, 0.67)**
Hispanic	0.87 (0.69, 1.10)	**0.70 (0.57, 0.87)**
Other/multiracial, non-Hispanic ^d^	0.96 (0.74, 1.23)	1.01 (0.80, 1.27)
**Caregiver Characteristics**		
**Maternal age (years)**		
>25	Reference	Reference
≤25	**0.76 (0.61, 0.94)**	0.89 (0.73, 1.10)
**Caregiver education**		
≥College degree	Reference	Reference
Some college or Associate’s	**0.64 (0.52, 0.78)**	0.85 (0.70, 1.03)
≤High school	0.77 (0.48, 1.22)	1.19 (0.76, 1.86)
**Childcare from someone other than parent or guardian for ≥10 h per week**		
Yes	Reference	Reference
No	0.86 (0.73, 1.02)	**0.83 (0.72, 0.96)**
**Household characteristics**		
**Household income ^e^**		
≥350% FPL	Reference	Reference
130% to <350% FPL	0.86 (0.70, 1.06)	0.99 (0.81, 1.20)
<130% FPL	0.95 (0.68, 1.33)	1.00 (0.76, 1.32)
**Food assistance in the past 12 months ^f^**		
No	Reference	Reference
Yes	0.83 (0.68, 1.02)	0.94 (0.77, 1.14)
**Food insufficiency the past 12 months ^g^**		
Low	Reference	Reference
Marginal	0.95 (0.77, 1.18)	0.80 (0.65, 1.00)
High	0.79 (0.49, 1.28)	0.81 (0.49, 1.32)
**No. family meals in the past week**		
Every day	Reference	Reference
4–6 days	**0.69 (0.57, 0.83)**	**0.70 (0.59, 0.83)**
0–3 days	**0.57 (0.46, 0.72)**	**0.45 (0.36, 0.55)**

Abbreviations: aOR, adjusted odds ratio; CI, confidence intervals; FPL, federal poverty level. ^‡^ The regression models included children with complete data (n = 15,816). ^a^ All variables in Table 2 were included in the multiple logistic regression models); reference outcome categories were daily fruit intake and daily vegetable intake; significant findings are bolded based on the 95% confidence intervals, which does not include 1. ^b^ Daily fruit was assessed by asking “During the past week, how many times did this child eat fruit? *Include any that were fresh, frozen, canned, or dried. Do not include juice.*” “Daily” was defined as responses of either 1 time, 2 times, or 3 or more times per day in the last week. ^c^ Daily vegetable was assessed by asking “During the past week, how many times did this child eat vegetables?” *Include any that were fresh, frozen, or canned. Do not include french fries, fried potatoes, or potato chips.*” “Daily” was defined as responses of either 1 time, 2 times, or 3 or more times per day in the last week. ^d^ This variable includes non-Hispanic American Indian or Alaska Native, Native Hawaiian and Other Pacific Islander, or Multiracial. ^e^ This variable represents household income as a percentage of the federal poverty level (FPL) by family composition. This variable was multiply-imputed according to NSCH guidelines to account for missing values. Categories reflect cut points for participation in SNAP and provide relatively equal sample sizes for each income group. The 2021 federal poverty guidelines can be found at https://aspe.hhs.gov/2021-poverty-guidelines#guidelines (accessed on 19 October 2023). ^f^ This variable reflects participation in SNAP or WIC or Free/Reduced Lunch benefits in the previous 12 months. ^g^ Food insufficiency was defined as “low” (could always afford to eat good nutritious meals), “marginal” (could always afford enough to eat but not always the kinds of foods we should eat), or “high” (sometimes or often could not afford enough to eat).

## Data Availability

Data supporting the results of this analysis are made available to the public by the US Census Bureau at https://www.census.gov/programs-surveys/nsch/data/datasets.html, accessed on 19 October 2023. Additionally, the analytic dataset used in this analysis is available upon request from the corresponding author.

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
