# Peer review of "Factors Associated with Daily Fruit and Vegetable Intakes among Children Aged 1–5 Years in the United States"

_nutrients, 2024, doi:10.3390/nu16050751_

Round 1
Reviewer 1 Report
Comments and Suggestions for Authors
The authors analyzed data from the 2021 National Survey of Children’s Health for 1–5-year-olds, the first survey year to include data about fruit and vegetable intake. Available data on children in this age range is limited. The aim was to describe child, caregiver and household characteristics associated with fruit and vegetable intake. Data confirm some previously identified trends with decreased fruit and vegetable intakes in non-Hispanic Black compared to non-Hispanic White children, but results question some previous associations made between race and ethnicity and household income impacting intake of fruits and vegetables. The manuscript is well-written although the quality of figure/tables could be improved to aid readability.
Major comment:
1. Figure 1 breaks down the data by age (1-5 years) as do Tables 1 and Table 2. There is a significant trend of declining fruit and vegetable intake progressively as children age during this 5 year period that is cited in “children characteristics” in Table 1, and mentioned in the results. While this was not the primary research question, the authors do present the data, but don’t’ give it adequate discussion. Please comment on this in the discussion as it seems to be a poignant and alarming statistic, especially considering that these children are still very young and do not eat on their own without a caregiver. Is this speculated to be due to the influence of media exposure as kids grow and gain access to TV, cell phones, tablets, and other media that they like f & v less? Have others cited this decrease in fruits and vegetables from infancy into toddlerhood? And does this continue into the next age groups (6-10 year olds, etc)? This is a compelling bit of data that should be addressed.
Minor comments:
2. Figures 1 and 2 could be made to look more professional and readable by using x- and y-axis lines, bolded text in some places, etc
3. Tables could also be restyled to improve readability, perhaps with strategic shading. If table spans more than one page in final form, please add headers to the continued portion.
Reviewer 2 Report
Comments and Suggestions for Authors
The paper titled “Factors Associated with Daily Fruit and Vegetable Intakes 2 among Children Aged 1–5 Years in the United States” is well-written and is timely.
The authors should mention in the text the potential impact of not eating enough food and vegetables in 1-5 years in more detail.
Would taking supplement in the form of vitamins help the child if he/she is not eating enough vegetables. Please comment on that.
Why are fried potatoes not considered as vegetable intake? Is it for the oil intake? Does that also hold for other vegetables? Does the fried broccoli or carrots consider as vegetable intake? Please mention it.
Is taking canned vegetables/fruits advisable? It may have preservatives or sugar. Please comment on that.
Do the authors think that physical activity and screen time play a part in vegetable and fruit intake?
Please comment on that.
